# Physiological demands and motion analysis of elite foil fencing

**Lindsay Bottoms**[1]*, **Rafael Tarragó**[2], **Daniel Muñiz**[1], **Diego Chaverri**[2], **Alfredo Irurtia**[2],
**Jorge Castizo-Olier**[2,3], **Marta Carrasco**[2], **Ferran A. Rodríguez**[2], **Xavier Iglesias**[2]

1 Centre for Research in Psychology and Sport Sciences, University of Hertfordshire, Hatfield, United
Kingdom, 2 Grup de Recerca en Ciències de l'Esport INEFC Barcelona (GRCEIB), Institut Nacional
d'Educació Física de Catalunya (INEFC), Universitat de Barcelona, Barcelona, Spain, 3 School of Health
Sciences, TecnoCampus, Pompeu Fabra University, Barcelona, Spain

* l.bottoms@herts.ac.uk

## Abstract

The aim of this study was to determine the physiological demands and motion analysis of a
simulated fencing competition. Eighteen fencers each completed 5 'poule' (5 touches) and
'direct elimination (DE)' (15 touches) fights. Expired gases were measured during 1 poule
and DE fight. Heart rate (HR), ratings of perceived exertion (RPE) and movement data were
recorded throughout all fights. Motion analysis was undertaken using the software LINCE
PLUS. Differences between poule and DE fights were determined by either a paired t-test or
a one-way repeated measures ANOVA. HR and RPE were significantly greater during DE
compared to poule (170 ±10 vs 163 ±13 beats·min$^{-1}$; P<0.05). A greater distance was cov-
ered during a DE fight compared to a poule fight (459.9 ± 117.7 m vs 162.6 ± 74.2 m;
P<0.05). The average values of $\dot{V}O_{2max}$ were 42.5 ±5.6 ml·kg$^{-1}$·min$^{-1}$ in men and 34.4 ±3.2
ml·kg$^{-1}$·min$^{-1}$ in women. Work-to-rest ratios reduced during the DE fights along with a lower
average speed and increased max speed (11.7 ± 2.8 km·h$^{-1}$ vs 9.6 ± 1.6 km·h$^{-1}$; P<0.05). In
conclusion, there is an increased physiological demand during a DE fight accompanied by a
reduction in average speed and decreased work-to-rest ratio.

doi.org/10.1371/journal.pone.0281600

Milano, ITALY

**Data Availability Statement:** Data relevant to this
study are available from https://doi.org/10.18745/
DS.26019 [Bottoms, L., Tarragó, R., Muñiz, D.,
Chaverri, D., Irurtia, A., Castizo-Olier, J., Carrasco,
M., Rodríguez, F.A. and Iglesias, X. (2023).

## Introduction

Fencing is an Olympic Sport and has 36 medals on offer at the Olympics. There are 3 different
weapons, foil, épée and sabre and each have specific tactical characteristics which make them
physiologically different [1]. Épée is the weapon which has been most researched, and this
involves the fencer trying to touch their opponent anywhere from head to toe and touches can be
scored by both fencers at the same time. Points are scored by only the tip of the sword contacting
the opponent. Both foil and sabre have a system of priority where the fencer who initiates the
attack has the right to score the point unless the opponent manages to successfully parry (a defen-
sive movement). Foil has a target area of just the torso and a point is scored with the tip of the
weapon whereas sabre a point is scored with any part of the weapon making contact from the
waist up. These differences mean that the tactics and work-to-rest ratios for each weapon are dif-
ferent resulting in different physiological responses. The present study focuses on foil fencing.

Physiological demands of elite foil fencing [Data set]. University of Hertfordshire.].

**Funding:** Professor Xavier Iglesias received funding from the Ministerio de Universidades (Spain) for mobility stays for Professors in foreign centers. The funders had no role in study design, data collection and analysis, decision to publish, or preparation of the manuscript. All other authors have no relevant financial or non-financial interests to disclose.

**Competing interests:** The authors have declared that no competing interests exist.

The duration of a competition for a fencer is very diverse. It depends on the type of competition and the fencer's final result, and can be from a few minutes, losing a direct elimination (DE) fight, as can happen in the Olympic Games [2], to a period of 9 to 11 hours that includes poule and DE fights [3]. Poule fights are defined as first to 5 touch fights during 3-minutes of fencing. If the 3 minutes are reached the winner is determined by most points won unless the fight is drawn, then up to an extra minute of fencing is completed. Fencers will compete in 4–6 poule fights in a round-robin format after which fencers are seeded for knockout DE fights. Direct elimination fights comprise of first to 15 touches during 3x3 minute bouts with 1 minute of rest between them. If scores are tied after the final 3-minute bout a 1-minute sudden death bout will determine the winner [2]. Fencers could potentially compete in up to 8 DE fights during a competition depending on athlete numbers in the competition. The average work-to-rest ratios for foil has been found to be ~1:3 [4] but the regulations [2] and the official duration times of the fights have changed a lot over the last 20 years since its publication.

There is limited research determining the physiological demands of fencing, with the majority of research taking place in épée and in simulated or laboratory environments with no competitive element [5–12]. Understanding the demands of fencing would allow coaches and practitioners to set training programmes to prepare athletes for competition by attempting to match training and conditioning sessions to competition demands. Only two apparent studies have reported heart rate (HR) responses for foil fencing which were found to be ~92.5% $HR_{max}$ in a poule fight and 96.5% for a DE fight with adolescent females [13] and 173 ±7 beats·min$^{-1}$ on average for a fight [14]. There is no data on oxygen consumption for foil but blood lactate was found to be 4.2 mmol·L$^{-1}$ during an official women's competition [14].

Previous research in foil has attempted to quantify movement patterns using time-motion analysis [13, 15, 16]. Movement patterns in foil fencing have been determined as low (walking and stationary) for 40–45% or moderate (engaged; movement on the piste during engagement such as before an attack or defensive movement) for ~50% with around 5–10% as high intensity (sustained or explosive attacking or defensive) movements [13, 15]. With the subjective and time-consuming nature of time-motion analysis, as well as technological advances analysing movement demands has shifted towards GPS/accelerometer-based systems which can provide immediate information such as speed, distance, accelerations etc. and have been validated for other intermittent sports [17]. Understanding the movement characteristics of fencing performance would allow coaches and practitioners to create a more in-depth picture of fencing performance and to create training programmes to match the demands of competition. There has been no previous research in foil fencing using tri-axial accelerometer-based systems. Therefore, the aim of this study was to describe the physiological demands and motion analysis of a simulated foil fencing competition, determining if there are differences according to the type of fight: poule vs direct elimination.

## Materials and methods

### Participants

Eighteen fencers (6 female, 4 left-handed) volunteered to participate in the study (characteristics can be seen in Table 1). All participants had at least 3 years' experience of training in foil fencing and had been invited by the Spanish Fencing Federation to take part in the study, they were free from injury, and undertook more than 4 weekly training sessions. Six males and four females were in the top ten Spanish senior or junior rankings at the time of the data collection. One male participant was a current medallist at the senior world championships. Participants provided written informed consent and ethical approval (protocol number: 12/2018/CEICGC) was obtained by the Clinical Research Ethics Committee of the Catalan Sports Ministry, in

**Table 1. Fencers characteristics: Age, body height (H), body mass (BM) and body mass index (BMI).**

|  | Male (n = 12) | Female (n = 6) | All (n = 18) |
|---|---|---|---|
| **Age (years)** | 20.0 ±5.8 | 21.0 ±3.9 | 20.3 ±5.1 |
| **H (cm)** | 178.4 ±8.5 | 163.2 ±4.0 | 173.3 ±10.3 |
| **BM (kg)** | 71.5 ±11.1 | 59.8 ±6.6 | 67.6 ±11.2 |
| **BMI (kg·m$^{-2}$)** | 22.4 ±2.3 | 22.4 ±2.1 | 22.4 ±2.2 |

Values are Mean ±SD

accordance with the principles outlined in the Declaration of Helsinki. Once consent was provided, participants were given an ID number for data collection and analyses. All participants were in a rested state prior to testing having refrained from alcohol and vigorous exercise in the 24 hours prior and caffeine and food ingestion in the 2 hours prior to testing. Furthermore, they arrived for the measurements after voiding their bladder and rectum [18].

## Study design

This study used an observational design where the fencers were divided in to 3 groups of 6 fencers of similar ability (determined by the coach). They then completed 'poule' (up to 5 touches) fights and 'DE' (up to 15 touches) fights with each of the fencers in their group. We used a counterbalanced design so that half of the fencers started with a 5-touch round and the other half with the 15-touch round (Fig 1). All testing was performed using the participants' own fencing equipment and participants were instructed to prepare as they would for competition. The experiment was carried out during the competitive period. The project was undertaken during a training camp of the Spanish national team at the high performance centre CAR Sant Cugat (Spain) in July 2018. One of the training sessions was dedicated to the undertaking of this project. All fencing was completed in an air-conditioned sports hall which was maintained at approximately 21˚C.

## Procedures

**Fencing protocol.** After a 10-min warm-up consisting of stretching, fencing movements and a 3-min fight, participants were assigned into their groups. The fencers completed 5 poule fights and 5 DE fights, in pairs, against the same opponent of similar fencing ability. The fights

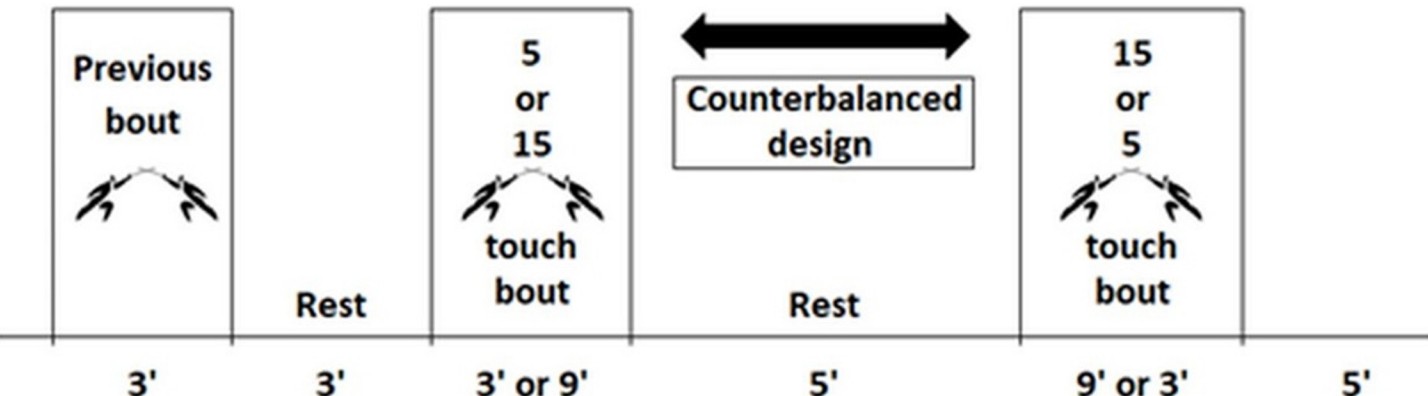

**Fig 1. Counterbalanced design so that half of the fencers will start with a 5-touch round and the other half with the 15-touch round.** Fight order with expired gases analysis is randomly assigned among participants.

were timed and refereed following the official fencing regulations [2]. The fencers had 5 to 10 minutes of rest between fights. Of the 5 pairs of fights (poule and DE), one was chosen at random, for each pair of fencers, for evaluation using the gas analyser.

**Anthropometric and bioelectrical impedance assessment.** Prior to the start of the fencing, anthropometric measurements were performed according to the standard criteria of The International Society for the Advancement of Kinanthropometry (ISAK) [19]. Participants were assessed wearing underwear. Body height (H) was assessed to the nearest 1 mm using a telescopic stadiometer (Seca 220®, Hamburg, Germany). Body mass (BM) was measured to the nearest 0.05 kg using a calibrated weighing scale (Seca 710®, Hamburg, Germany). Body mass index (BMI) was calculated as $kg \cdot m^{-2}$. Wingspan was measured to the nearest 1 mm using a wingspan measurer (RealmetBcn, Barcelona, Spain).

Bioelectrical impedance analysis (BIA) measurements were performed using the InBody 720 (Biospace Co., California, USA). Participants were asked to cleanse their hands and feet with antibacterial tissues before touching the electrodes. They stood erect with bare feet on the device with both arms extended and abducted from the trunk, with all four extremities in contact with the electrodes. The instrument measured impedance at varying frequencies (1, 5, 50, 250, 500 and 1,000 kHz) across the legs, arms, and trunk in order to estimate body fat mass (% FM), body skeletal muscle mass (%MM), right and left arms lean mass, and right and left legs lean mass. Extremities dominancy was considered as follows: the dominant arm was the foil-wielding arm, and the dominant leg was the ipsilateral leg of the dominant arm.

**Blood measures.** Blood lactate concentration was measured by taking a 0.3μl capillary blood sample from an earlobe. Blood lactate was then analysed using a handheld analyser (Lactate Pro 2, Arkray, Japan). Blood lactate was measured at the following time points: baseline, after the last poule fight and after the last DE fight. The maximum value of the samples measured in minutes 1 and 3 was taken as maximum, according to previous studies [14]. Blood glucose was measured at baseline and after the last bout of fencing by taking another 0.3μl blood sample from the earlobe and analysed using Accu-Chek Mobile blood glucose monitoring system.

**Heart rate and movement tracking.** Participants were fitted with a heart rate monitor and athlete tracking system (Polar Team Pro 2, Polar Electro, Kempele, Finland). Heart rate and movement data were tracked continuously throughout all fencing using an accelerometer, gyroscope and digital compass system recording at 200Hz. Average heart rate ($HR_{av}$) and maximum heart rate ($HR_{max}$) during each poule fight and DE fight were recorded. A percentage of $HR_{max}$ was determined using the standard age predicted HRmax (220-age). Distance covered and peak speed were analysed for all fights. Ratings of perceived exertion (RPE) were recorded using the Borg 1–10 category scale [20].

Motion analysis was undertaken by video recording all fights using a digital Sony HDR-XR550VE. A well-trained observer and expert in previous analyses [21] made the recordings. It was registered using the software LINCE PLUS [22, 23]. The definitions of the motion analysis variables are: i) Total fight time (s): The time the fight started to the time it ended, including any stoppage time; ii) Active fight time (s): The time the fencers were fighting during the fight; iii) Rest fight time (s): The time between when the referee called halt to when they called a new start. Usually this is between referee decisions. During DE fights, this includes the 1-minute rest between 3-minute periods; and iv) Work-to-rest ratio: This is the ratio between active fight time and rest time during the fight.

Work-to-rest times (total fight time, active fight time, rest time and work-to-rest ratio) were calculated using a poule fight and a DE fight on each fencer. Distance and speed measures (total distance, average speed and max speed) were calculated based on 5 poule fights and another 5 DE fights for each fencer.

**Core temperature.** Core temperature was measured throughout using an ingestible telemetric core temperature pill (CorTemp, HQ Inc., Palmetto, FL, USA) and analysed for baseline, pre and post poule fights and pre and post DE fights. The participants were provided the core temperature pills to ingest 2 hours prior to the start of the poule fights. When taking measurements, the data recorder was held 2–3 cm behind the participants back for all core temperature measurements as per manufacturer guidelines.

**Expired gases and energy expenditure.** Expired gases were measured during 1 poule fight and 1 DE fight with the same opponent using a portable telemetric breath-by-breath gas analyser (Cosmed K4 b$^2$, Italy) connected to an oronasal Hans-Rudolph 7400 mask (Hans Rudolph Inc., Shawnee, Kansas, USA) that fitted comfortably under the participant's own fencing mask. Expired gases were analysed for oxygen uptake ($\dot{V}O_2$), carbon dioxide production ($VCO_2$) and respiratory exchange ratio (RER). Indirect calorimetry was used to calculate rate of energy expenditure (EE; kcal·min$^{-1}$) using stoichiometric equations specifically developed for exercise at moderate to high intensity using the following equation [24]:

$$\text{Energy Expenditure for high intensity exercise (kcal} \cdot \text{min}^{-1}) = \left[(0.550 \cdot \dot{V}CO_2) - (4.471 \cdot \dot{V}O_2)\right]$$

Prior to each test, analysers were calibrated with gases of known concentration and the linearity of the gas meter was checked by a 3-litre calibration syringe according to manufacturer's guidelines. All gas analysis variables were analysed for each subject during 1 poule fight and 1 DE fight (Fig 1).

## Statistical analysis

Data are presented in tables as means ±standard deviations for all participants as well as for males and females. Data were analysed using a statistical software package (SPSS version 27, IBM, Armonk, NY, USA). Data were checked for normality using the Shapiro-Wilk test. Paired samples t-test analyses were undertaken on the anthropometric data between the dominant and non-dominant side as well as on physiological responses and motion analysis variables between poule and DE fights. Effect sizes (ES) for differences between dominant and non-dominant were calculated using Cohen's d [25] and considered to be trivial (ES < 0.20), small (0.21–0.60), moderate (0.61–1.20), large (1.21–2.00), or very large (ES >2.00) [26]. A one-way repeated measures analysis of variance (ANOVA) was undertaken to compare the blood lactate response at baseline, end of the poule fights and end of the DE fights with Bonferonni corrected post hoc analyses to determine where the differences existed. A one-way repeated measures analysis of variance (ANOVA) was undertaken for core temperature between baseline, pre and post poule and pre and post DE fights. Bonferonni corrected post hoc analyses were undertaken to determine where the differences existed. Significance was accepted as P<0.05.

## Results

As can be seen from Table 2, wingspan for all participants was 174.5 ±11.0 cm. Participants reported a fat mass % of 14.8 ±6.9 and muscle mass % of 48.0 ±4.4. When broken down into limbs and compared between dominant and non-dominant sides, lean mass was significantly less in the non-dominant arm ($T_{(17)}$ = 9.62; P<0.001, $d$ = 2.27) and the same for the leg ($T_{(17)}$ = 2.83; P = 0.012, $d$ = 0.67).

The fencers presented baseline lactate values of 1.3 ±0.2 mmol·L$^{-1}$. After the fights the values increased in both poule (2.3 ±0.7 mmol·L$^{-1}$) and DE (2.4 ±1.1 mmol·L$^{-1}$). A one-way repeated measures ANOVA was performed. The assumption of sphericity was violated, as

**Table 2. Wingspan and body composition for all fencers.**

|  | Male (n = 12) | Female (n = 6) | All (n = 18) |
|---|---|---|---|
| **Wingspan (cm)** | 180.0 ± 8.8 | 163.6 ± 4.5 | 174.5 ± 11.0 |
| **Body Fat Mass (%)** | 11.3 ± 4.7 | 21.6 ± 5.2 | 14.8 ± 6.9 |
| **Body Skeletal Muscle Mass (%)** | 50.3 ± 2.9 | 43.4 ± 2.8 | 48.0 ± 4.4 |
| **Lean Mass Dominant Arm (kg)** | 3.6 ± 0.7 | 2.4 ± 0.3 | 3.2 ± 0.8 |
| **Lean Mass Non-Dominant Arm (kg)** | 3.3 ± 0.7 | 2.2 ± 0.2 | 2.9 ± 0.8* |
| **Lean Mass Dominant Leg (kg)** | 10.1 ± 1.7 | 7.1 ± 0.8 | 9.1 ± 2.0 |
| **Lean Mass Non-Dominant Leg (kg)** | 10.0 ± 1.7 | 7.1 ± 0.8 | 9.0 ± 2.0* |

Values are Mean ±SD;

* denotes significant difference from dominant side

assessed by Mauchly's test of sphericity, $X^2_{(2)}$ = 9.37, P<0.01. Therefore, a Greenhouse-Geisser correction was applied ($\varepsilon$ = 0.693). The different time points elicited statistically significant changes in blood lactate concentration, $F_{(1.39, 23.56)}$ = 23.06; P<0.001, partial $\eta^2$ = 0.576. Post hoc analysis with a Bonferonni adjustment revealed blood lactate concentration was statistically increased from baseline to end of the poule fights 1.2 mmol·L$^{-1}$ (95% CI, 0.6 to 1.7 mmol·L$^{-1}$) and remained similar at the end of the DE fights 0.06 mmol·L$^{-1}$ (95% CI, -0.26 to 0.38 mmol·L$^{-1}$).

With regards to blood glucose, there was a tendency for a higher blood glucose at the end of all fencing (5.7 ±0.8 mmol·L$^{-1}$) compared to baseline (5.3 ±0.6 mmol·L$^{-1}$), with an increase ($T_{(17)}$ = -2.08, P = 0.053, $d$ = 0.49) of 0.44 mmol·L$^{-1}$ (95% CI, 0.21 to 0.88 mmol·L$^{-1}$).

Table 3 presents HR, RPE and gas analysis variables. As can be seen in Table 3, $HR_{av}$ and $HR_{max}$ were significantly greater during the DE fights than the poule fights (P<0.05). Subsequently, fencers RPE was greater during DE fights (5.6 ±1.6) compared to poule fights (3.7 ±1.3), a statistically significant difference of 1.9 ±1.8 (95% CI, 1.0 to 2.8), $T_{(17)}$ = -4.42; P<0.001, $d$ = -1.04.

We calculated the HR% relative to the age-predicted $HR_{max}$ in each participant. The global values show how the foil fencers in the study performed the poule bouts at a mean intensity of 81.6 ±6.5%, while in the DE bouts they were 85.3 ±4.9%. The male fencers presented mean values of 81.2 ±5.7% in poule and 84.1 ±5.1% in DE, while the women obtained 82.4 ±8.4% and 87.6 ± 3.8%, respectively.

There were no differences in absolute $\dot{V}O_{2max}$ or EE between poule and DE fights (P>0.05). The average values of relative $\dot{V}O_{2max}$ in the DE were 42.5 ±5.6 ml·kg$^{-1}$·min$^{-1}$ in men and 34.4 ±3.2 ml·kg$^{-1}$·min$^{-1}$ in women. However, there was a higher RER in a DE fight (0.98 ±0.04) compared to a poule fight (0.94 ±0.09), a statistically significant increase of 0.04 ±0.08 (95% CI, 0.00 to 0.08), $T_{(17)}$ = 2.12; P = 0.048, $d$ = -0.50.

The core temperature measurements were taken during the simulated competition at various time points. Baseline values were 37.3 ±0.4˚C. After warm-up and before the poule fights (pre poule) the temperature was 37.8 ±0.5˚C, and at the end (post poule) 38.0 ±0.5˚C. In the DE fights, the temperature was 37.7 ±0.3˚C in the pre DE and 38.2 ±0.4˚C in the post DE fights. A one-way repeated measures ANOVA was conducted to determine whether there was a statistically significant difference in core temperature at different time points. The different time points elicited statistically significant differences in core temperature, $F_{(4, 68)}$ = 22.28; P<0.001, partial $\eta^2$ = 0.567. Post hoc analysis with a Bonferonni adjustment revealed core temperature statistically increased from baseline to pre poule 0.49˚C (95% CI, 0.14 to 0.85˚C), post poule 0.70˚C (95% CI, 0.37 to 1.04˚C), pre DE 0.39˚C (95% CI, 0.15 to 0.64˚C) and post DE

**Table 3. Heart rate average (HR$_{av}$), maximum heart rate (HR$_{max}$), ratings of perceived exertion (RPE), maximum oxygen uptake (VO$_{2\ max}$), respiratory exchange ratio (RER) and energy expenditure (EE) for all fencers.**

| | Poule | Direct Elimination | Effect Size |
|---|---|---|---|
| **HR$_{av}$ (beats·min$^{-1}$)** | | | |
| *All (n = 18)* | 163 ±13 | 170 ±10* | -0.82 |
| *Male (n = 12)* | 162 ± 12 | 168 ±12 | |
| *Female (n = 6)* | 164 ±16 | 174 ±6 | |
| **HR$_{max}$ (beats·min$^{-1}$)** | | | |
| *All (n = 18)* | 178 ±11 | 185 ±11* | -1.03 |
| *Male (n = 12)* | 177 ±13 | 184 ±13 | |
| *Female (n = 6)* | 181 ±7 | 187 ±6 | |
| **RPE** | | | |
| *All (n = 18)* | 3.7 ±1.2 | 5.6 ±1.6* | -1.04 |
| *Male (n = 12)* | 4.0 ±1.2 | 5.4 ±1.5 | |
| *Female (n = 6)* | 3.2 ±1.2 | 6.0 ±2.0 | |
| **VO$_{2\ max}$ (L·min$^{-1}$)** | | | |
| *All (n = 18)* | 2.61 ±0.61 | 2.71 ±0.69 | -0.29 |
| *Male (n = 12)* | 2.83 ±0.56 | 3.03 ±0.60 | |
| *Female (n = 6)* | 2.16 ±0.43 | 2.05 ±0.28 | |
| **RER** | | | |
| *All (n = 18)* | 0.94 ±0.09 | 0.98 ±0.04* | -0.50 |
| *Male (n = 12)* | 0.97 ±0.08 | 0.99 ±0.04 | |
| *Female (n = 6)* | 0.90 ±0.08 | 0.97 ±0.04 | |
| **EE (kcal·min$^{-1}$)** | | | |
| *All (n = 18)* | 10.3 ±2.4 | 10.6 ±2.7 | -0.25 |
| *Male (n = 12)* | 11.2 ±2.2 | 11.9 ±2.3 | |
| *Female (n = 8)* | 8.6 ±1.1 | 8.1 ±1.1 | |

Values are Mean ±SD;

* denotes significant difference from poule

0.86˚C (95% CI, 0.50 to 1.22˚C). Post DE core temperature was greater than pre poule 0.37˚C (95% CI, 0.04 to 0.71˚C) and finally post DE core temperature was greater than pre DE temperature 0.47˚C (95% CI, 0.22 to 0.71˚C).

Table 4 illustrates the various motion analysis variables and the differences between poule and DE fights. As can be seen from Table 4 total fight time was greater during the DE fight compared to the poule fight, with a statistically significant difference of 344.2 s (95% CI, 251.9 to 436.5 s), $T_{(17)} = 7.87$, P<0.001, $d$ = -1.85. There was also longer active fight time during the DE fight compared to the poule fight, with a statistically significant difference of 120.9 s (95% CI, 88.6 to 153.3s), $T_{(17)} = 7.89$, P<0.001, $d$ = 1.81. This was accompanied with a significantly greater total rest time during the DE fight compared to the poule fight, with a significant difference of 245.1 s (95% CI, 147.9 to 231.9 s), $T_{(17)} = 9.45$, P<0.001, $d$ = -2.23. This led to a significantly lower work-to-rest ratio during the DE fights compared to the poule fights, with a significant difference of 0.15 (95% CI, 0.02 to 0.28), $T_{(17)} = 2.42$, P = 0.028, $d$ = -0.59.

The fencers covered a greater distance during the DE fights compared to the poule fights, with a significant difference of 297.2 m (95% CI, 240.3 to 354.1 m), $T_{(17)} = 11.0$, P<0.001, $d$ = -2.57. Maximum speed was greater during a DE fight compared to a poule fight, with a significant difference of 2.19 m.s$^{-1}$ (95% CI, 0.72 to 3.66 m.s$^{-1}$), $T_{(17)} = 3.15$, P = 0.006, $d$ = -0.74.

**Table 4. Motion analysis variables for both poule and direct elimination fights.**

| | Poule | Direct Elimination | Effect Size |
|---|---|---|---|
| **Total fight time (s)** | | | |
| *All (n = 18)* | 173 ±61.9 | 554.4 ±209.3* | -1.85 |
| *Male (n = 12)* | 169.7 ±58.1 | 470.6 ±210.3 | |
| *Female (n = 6)* | 241.4 ±80.4 | 672.3 ±152.3 | |
| **Active fight time (s)** | | | |
| *All (n = 18)* | 78.3 ±37.1 | 197.9 ±77.5* | -1.86 |
| *Male (n = 12)* | 77.3 ±38.3 | 188.9 ±72.8 | |
| *Female (n = 6)* | 102.9 ±43.8 | 242.4 ±76.0 | |
| **Total rest time (s)** | | | |
| *All (n = 18)* | 91.4 ±36.8 | 356.5 ±140.0* | -2.23 |
| *Male (n = 12)* | 76.0 ±27.6 | 298.0 ±130.8 | |
| *Female (n = 6)* | 138.6 ±42.8 | 429.9 ±86.1 | |
| **Work-to-rest ratio** | | | |
| *All (n = 18)* | 1.0:1.2 | 1.0:1.4* | -0.59 |
| *Male (n = 12)* | 1.0:1.1 | 1.0:1.4 | |
| *Female (n = 6)* | 1.0:1.5 | 1.0:1.5 | |
| **Total Distance (m)** | | | |
| *All (n = 18)* | 162.6 ±74.2 | 459.9 ±117.7* | -2.60 |
| *Male (n = 12)* | 147.0 ±57.5 | 454.2 ±79.6 | |
| *Female (n = 6)* | 193.8 ±98.4 | 471.2 ±181.4 | |
| **Average Speed (km·h$^{-1}$)** | | | |
| *All (n = 18)* | 3.2 ±0.7 | 2.9 ±0.8* | -0.63 |
| *Male (n = 12)* | 3.5 ±0.6 | 3.1 ±0.9 | |
| *Female (n = 6)* | 2.7 ±0.62 | 2.5 ±0.38 | |
| **Max Speed (km·h$^{-1}$)** | | | |
| *All (n = 18)* | 9.6 ±1.6 | 11.7 ±2.8* | -0.73 |
| *Male (n = 12)* | 9.6 ±1.9 | 12.7 ±2.8 | |
| *Female (n = 6)* | 9.4 ±1.0 | 9.9 ±1.5 | |

Values are Mean ±SD;

* denotes significant difference from poule

## Discussion

The aim of this study was to simulate a foil fencing competition and provide descriptive data of different physiological variables as well as determining movement through both motion analysis and a tri-axial accelerometer-based system. Furthermore, determining if there are differences according to the type of fight: poule vs direct elimination. This is the first study to provide such a comprehensive physiological and motion analysis overview of a simulated men's and women's foil fencing competition.

### Anthropometric and bioelectrical impedance assessment

The study participants were selected by the Spanish national foil fencing team for a training camp in which the measurements were made. The anthropometric measurements illustrated the characteristic muscular asymmetries of fencers already described in the literature [3, 9, 27–29].

In relation to the BMI values of our study, the values obtained are very similar in men (22.4 ±2.3 kg·m$^{-2}$) and women (22.4 ±2.1 kg·m$^{-2}$). Our values in men are very close to those recorded in the literature, but in the female sample they differ moderately with lower values

[30] (male 22.21±2.93 kg·m$^{-2}$ and female 20.33 ±5.98 kg·m$^{-2}$ [31]; male 22.9±1.3 kg·m$^{-2}$ and female 19.9±3.3 kg·m$^{-2}$) or higher [32] (male 23.2 ±9.2 kg·m$^{-2}$ and female 22.5 ±9.7 kg·m$^{-2}$).

The BMI values of our sample of elite male foil fencers is very similar to that of 14 elite épée athletes from the Czech Fencing Federation, who presented values of 22.86 ±2.97 kg·m$^{-2}$ [33]. Additionally, percentage fat mass values are similar between the studies with our fencers having 11.3 ±4.7% and the Czech fencers 11.09 ±4.91%.

## Heart rate

The current study produced an average HR of 163 ±13 beats.min$^{-1}$ (81.6 ±6.5% age-predicted HR$_{max}$) in the poule fights and 170 ±10 beats.min$^{-1}$ (85.3 ±4.9% age-predicted HR$_{max}$) in the DE fights. The main findings highlight a clear increase in physiological strain during the direct elimination fights compared to the earlier poule fights as demonstrated by the elevated HR, RPE and RER.

In a previous study by Iglesias & Rodríguez (1995) 182 fights in an official competition were analyzed for 7 male épée fencers and 6 female foil fencers. Heart rate was higher in women's foil (173 ± 3 beats·min$^{-1}$) than in men's épée (166 ± 3 beats·min$^{-1}$). The fencing competition system has significantly changed during the 30 years since this study, however, the HR values during the DE bouts in the present study for our female foil sample are very similar (174 ± 6 beats·min$^{-1}$) to those reported in the aforementioned study and higher than those reported by our male foil sample (168 ±12 beats·min$^{-1}$). These values in men's foil are similar to those obtained in previous studies in official competition (166 ± 3 beats·min$^{-1}$) [14] and in simulated men's épée competition. (169 ± 14 beats·min$^{-1}$) [34]. In our study, the male fencers presented HR average values of 81.2 ±5.7% in poule and 84.1 ±5.1% in DE of age-predicted HR$_{max}$, which is lower than that described in male épée fencers (86.3% and 86.5% age-predicted HR$_{max}$) [34]. The values in our female sample represent lower levels of intensity (82.4 ± 8.4% poule and 87.6 ± 3.8% DE of age-predicted HR$_{max}$) than those presented in a study with adolescent foil girls (92.5% poule and 96.5% DE of HR$_{max}$) [13]. The difference in intensity is most likely due to our study having a more experienced sample of fencers with a much higher mean age (21.0 ± 3.9 years) than the sample of adolescent girls (14.3 ± 1.2 years).

The use of different muscle groups during the fights (legs in movements, arm in the use of the weapon and trunk in many actions), the necessary intensity of the successful actions and the emotional strain of competition mean that the physiological demands in the fights are high, even considering the variability of the effort throughout the competition. Previous research has shown that HR in an official fencing competition was above ventilatory threshold 2 40.7% of the time, and 39.2% between ventilatory thresholds 1 and 2 [10].

The perceived exertion presented by foil fencers in fights is similar to that described in épée fencers [34] with values between moderate and hard.

## Expired gases

The average values of relative V̇O$_{2max}$ in the DE were 42.5 ±5.6 ml·kg$^{-1}$·min$^{-1}$ for the men and 34.4 ±3.2 ml·kg$^{-1}$·min$^{-1}$ for the women. Oxygen consumption was similar between poule and DE fights in the current study.

Our research group carried out previous work on the analysis of oxygen consumption in fencing. In a sample of English épée athletes [34], V̇O$_{2max}$ values were also similar between poule and DE fights (49.1 vs. 51.2 ml·kg$^{-1}$·min$^{-1}$). Previously estimated maximum oxygen consumption in an official fencing competition for men's épée fights were calculated to be 53.9 ± 4.4 ml·kg$^{-1}$·min$^{-1}$, while in women's foil it was estimated to be 39.6 ± 7.2 ml·kg$^{-1}$·min$^{-1}$ [10]. Both values are higher than those reported in épée [34] and also the male and female

fencers of the present foil study. However, these values could have been overestimated, as suggested by a subsequent validation study [11]. In another recent study [7], in men's épée, with club-level athletes, similar values were observed in fencing fights to those described previously (44.2 ±7.8 ml·kg$^{-1}$·min$^{-1}$).

Within the current study energy expenditure was shown to be 10.3 ±2.4 kcal·min$^{-1}$ and 10.6 ±2.7 kcal·min$^{-1}$ for poule and DE foil fights respectively, which is similar to those previously reported of 12.0 kcal·min$^{-1}$ in épée fencers [5]. These values are lower in relation to the estimated energy expenditure in official competition (15.4 vs. 12.3 kcal·min$^{-1}$ respectively) [11], but, as mentioned previously, these values were likely overestimated.

## Blood measures

The results of our study show a low demand for lactate metabolism both in poule (2.3 ±0.7 mmol·L$^{-1}$) and in DE fights (2.4 ±1.1 mmol·L$^{-1}$). These values, in simulated foil competition, are lower than the values recorded in an official women's foil competition (4.2 ± 0.9 mmol·L$^{-1}$) and men's épée (3.2 ± 0.7 mmol·L$^{-1}$). However, they are consistent with previous studies in female épée (~2.8 mmol·L$^{-1}$) [5]. In a similar study, in men's épée, lactate values with greater variability were recorded depending on the stage of the simulated competition (~2.08 to 4.54 mmol·L$^{-1}$) [34].

The results of this study and previous research highlights the importance of the alactic energy systems during fencing [5, 35] due to the lower blood lactate concentration observed during fencing fights. However, the repetition of many actions, of short duration, high intensity, and short recovery times, causes an increase in energy to be supplied from aerobic sources [36, 37].

Analysing these values, together with the cardiorespiratory results, we point out the relevance of cardiovascular demands and the prevalence of aerobic metabolism during fencing tournaments, as well as the relatively weak activation of glycolytic metabolism.

## Core temperature

In relation to core temperature, our study showed that temperature increases with fencing fights. We observed how all the measurements made during the simulated competition were higher than those collected at baseline. Likewise, post DE temperature was higher than pre poule 0.37˚C (95% CI, 0.04 to 0.71˚C) and pre DE 0.47˚C (95% CI, 0.22 at 0.71˚C).

These temperature values in foil are similar to those presented by épée fencers with pre fight increases in relation to post fight (37.65 vs 38.06˚C respectively). In épée fencers, a greater difference in temperature could also be observed in DE fights in relation to poule fights [34], which coincides with our values, although in our case there was not enough statistical power to confirm it. This could probably be due to the fact that in the épée study the two phases of consecutive fights were carried out (first all the poule and later those of DE) and the temperature could have been increasing as the global competition progressed, while in our foil study the order of fights was counterbalanced (Fig 1).

The increase in environmental temperature and humidity influences HR which affects energy expenditure [38]. Fencing is a sport in which, for safety reasons, the clothing is very thick to protect the whole body and includes wearing a mask to protect the head. This thick protective clothing could impact upon their ability to dissipate heat effectively, especially evaporative heat loss mechanisms [34, 39, 40]. Perhaps specific studies should be carried out to check, in similar environmental and sample conditions, the differences existing in foil and sabre fencers when wearing a conductive fabric jacket in addition to the clothing worn in épée.

## Motion analysis

This is the first study to determine the movement demands of foil fencing performance using a tri-axial accelerometer based system. The mean distance covered during the Poule and DE fights during this study were 162.6 ±74.2m and 459.9 ±117.7m respectively. These are lower than previous research observed in épée fencing [34]. The standard deviations are relatively high which shows the varying nature of foil fencing performance whereby the demand placed upon the body could be largely determined by the individual fight i.e. attacking vs defensive opponent. There was a significantly greater fight time during a DE fight compared to a poule fight which was accompanied by both an increase in active fight time and total rest. Interestingly, although average speed was lower during the DE fight compared to the poule there was an increase in maximum speed. The increased duration and higher target score of 15 means the fencers changed their tactics and rhythm of fencing compared to their actions in the earlier stages of a fight thus reducing the average speed slightly. This pattern was also seen in épée research with lower average speed in a DE fight and higher maximum speed [34].

The registered work-rest ratio values (1:1.4) are similar to those described by Wylde et al. [15] (1:3), lower than those registered in épée (0.8:0.9; [5] 1:1.0, [22] but higher than those described in sabre (1:6) [41]. In this case, the conventional weapons, in which arbitration decision is required, show higher values of rest than the épée, where the transitions without arbitration judgment are lower.

The greater number of touches that are required in DE fights compared to poule fights (15 vs 5) could condition a different tactical behaviour with fencers reducing their work-to-rest ratio, reducing their average speed and increasing their max speed.

## Conclusion

The study shows a physiological and motion profile of simulated men's and women's foil fencing competition. Low blood lactate levels and moderately high heart rate and oxygen uptake values, showing an important dependence on the aerobic and alactic energy system. Differences between poule fights and direct elimination fights are confirmed, with an increase in physiological demands (heart rate, ratings of perceived exertion and respiratory exchange ratio), which are accompanied by a lower work-rest ratio, lower average speed, and higher maximum speed during direct elimination fights.

## Acknowledgments

The authors would like to thank Martí Ferret and Víctor Aparicio for their assistance with data collection.

## Author Contributions

**Conceptualization:** Lindsay Bottoms, Rafael Tarragó, Diego Chaverri, Ferran A. Rodríguez, Xavier Iglesias.

**Data curation:** Lindsay Bottoms, Rafael Tarragó, Diego Chaverri, Alfredo Irurtia, Xavier Iglesias.

**Formal analysis:** Lindsay Bottoms, Daniel Muñiz, Xavier Iglesias.

**Investigation:** Lindsay Bottoms, Rafael Tarragó, Daniel Muñiz, Diego Chaverri, Alfredo Irurtia, Jorge Castizo-Olier, Marta Carrasco, Ferran A. Rodríguez, Xavier Iglesias.

**Methodology:** Lindsay Bottoms, Diego Chaverri, Alfredo Irurtia, Jorge Castizo-Olier, Marta Carrasco, Ferran A. Rodríguez, Xavier Iglesias.

**Project administration:** Lindsay Bottoms, Rafael Tarragó, Daniel Muñiz, Alfredo Irurtia, Jorge Castizo-Olier, Marta Carrasco, Ferran A. Rodríguez, Xavier Iglesias.

**Resources:** Diego Chaverri, Ferran A. Rodríguez, Xavier Iglesias.

**Software:** Xavier Iglesias.

**Supervision:** Lindsay Bottoms, Ferran A. Rodríguez, Xavier Iglesias.

**Writing – original draft:** Lindsay Bottoms, Xavier Iglesias.

**Writing – review & editing:** Rafael Tarragó, Daniel Muñiz, Diego Chaverri, Alfredo Irurtia, Jorge Castizo-Olier, Marta Carrasco, Ferran A. Rodríguez, Xavier Iglesias.

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
