## [Decision Letter · Decision Letter 0]

10 Jan 2023

PONE-D-22-26762Physiological demands and motion analysis of elite foil fencingPLOS ONE

Dear Dr. Bottoms,

Thank you for submitting your manuscript to PLOS ONE. After careful consideration, we feel that it has merit but does not fully meet PLOS ONE’s publication criteria as it currently stands. Therefore, we invite you to submit a revised version of the manuscript that addresses the points raised during the review process.

ACADEMIC EDITOR:Dear Authors,your manuscript has been reviewed by an expert in the field, which found some minor issues you should consider while revising the article. Please submit your revised manuscript by Feb 24 2023 11:59PM. If you will need more time than this to complete your revisions, please reply to this message or contact the journal office at plosone@plos.org. Please include the following items when submitting your revised manuscript:A rebuttal letter that responds to each point raised by the academic editor and reviewer(s). You should upload this letter as a separate file labeled 'Response to Reviewers'.A marked-up copy of your manuscript that highlights changes made to the original version. You should upload this as a separate file labeled 'Revised Manuscript with Track Changes'.An unmarked version of your revised paper without tracked changes. You should upload this as a separate file labeled 'Manuscript'.If applicable, we recommend that you deposit your laboratory protocols in protocols.io to enhance the reproducibility of your results. Protocols.io assigns your protocol its own identifier (DOI) so that it can be cited independently in the future. For instructions see: https://journals.plos.org/plosone/s/submission-guidelines#loc-laboratory-protocols. Additionally, PLOS ONE offers an option for publishing peer-reviewed Lab Protocol articles, which describe protocols hosted on protocols.io. Read more information on sharing protocols at https://plos.org/protocols?utm_medium=editorial-email&utm_source=authorletters&utm_campaign=protocols.

We look forward to receiving your revised manuscript.

Kind regards,

Emiliano Cè

Academic Editor

PLOS ONE

Journal Requirements:

We appreciate the funding received from the Ministerio de Universidades (Spain) for mobility stays for professors and researchers in foreign centers of higher education and research, the support of the Institut Nacional d’Educació Física de Catalunya (INEFC), Spanish Fencing Federation and CAR Sant Cugat.

The authors received no specific funding for this work.

Reviewers' comments:

Reviewer's Responses to Questions

**Comments to the Author**

1. Is the manuscript technically sound, and do the data support the conclusions?

Reviewer #1: Yes

2. Has the statistical analysis been performed appropriately and rigorously? 

Reviewer #1: Yes

3. Have the authors made all data underlying the findings in their manuscript fully available?

Reviewer #1: Yes

4. Is the manuscript presented in an intelligible fashion and written in standard English?

Reviewer #1: Yes

5. Review Comments to the Author

Reviewer #1: L43. Please provide a reference that fencing with foil, epee or sabre has different physiological demands and clarify that it is the fencing with different weapons that may induce different physiological demands.

L121. Please reconsider whether it was allocated and not randomized.

Ls 138-141. There is repeat here of info also communicated partly in Ls 118-123.

L143. Please reconsider “The order of the fights were randomised and counterbalanced, to minimise the effect of fatigue”. It is not clear how starting with the most demanding fencing would minimize fatigue.

L154. I suggest to change BM·H-2 with “kg·m-2”.

L171. The statement “The maximum value of the samples measured in minutes 1 and 3 was calculated, according to previous studies” is unclear why you need calculation. Please clarify.

L211. The dot on the first E in EE can be removed. Do this throughout the manuscript.

L255. Please check “0.06 mmol∙L-1 (95% CI, 0.26 to 0.38 mmol∙L-1)”. Is the 95% CI for the change of 0.06?

L265. Please provide the age-predicted equation in the methods.

6. PLOS authors have the option to publish the peer review history of their article (what does this mean?). If published, this will include your full peer review and any attached files.

Reviewer #1: **Yes: **Mark Willems

---

## [Author Response · Author response to Decision Letter 0]

16 Jan 2023

Thank you for reviewing our paper. We have gone through the feedback and tried to make the amendments suggested. Hopefully this has improved the manuscript. 

Reviewer #1: L43. Please provide a reference that fencing with foil, epee or sabre has different physiological demands and clarify that it is the fencing with different weapons that may induce different physiological demands.

Response: Thank you for your comment, we have now included a reference and added the word tactical to clarify why they are different. 

Reviewer: L121. Please reconsider whether it was allocated and not randomized.

Response: We have deleted randomised and called it counterbalanced. 

Reviewer: Ls 138-141. There is repeat here of info also communicated partly in Ls 118-123.

Response: Thank you, we have now deleted the repeated information. 

Reviewer: L143. Please reconsider “The order of the fights were randomised and counterbalanced, to minimise the effect of fatigue”. It is not clear how starting with the most demanding fencing would minimize fatigue.

Response: Thank you for the comment, to avoid confusion we have deleted the sentence. 

Reviewer: L154. I suggest to change BM·H-2 with “kg·m-2”.

Response: Thank you, we have now changed it to the recommended. 

Reviewer: L171. The statement “The maximum value of the samples measured in minutes 1 and 3 was calculated, according to previous studies” is unclear why you need calculation. Please clarify.

Response: We have amended the wording to “the maximum value of the samples measured in minutes 1 and 3 was taken as maximum, according to previous studies”

Reviewer: L211. The dot on the first E in EE can be removed. Do this throughout the manuscript.

Response: We have now removed the dot throughout. 

Reviewer: L255. Please check “0.06 mmol∙L-1 (95% CI, 0.26 to 0.38 mmol∙L-1)”. Is the 95% CI for the change of 0.06?

Response: Thanks for the query, I have gone and double checked the SPSS output and that is correct, although I missed the minus sign. The difference is 0.06 and it is the 95% CI for the difference in the pairwise comparisons. Hopefully it makes sense with the minus sign now. Apologies for missing it. 

Reviewer: L265. Please provide the age-predicted equation in the methods.

Response: Thank you for spotting this, we have now included the equation in the methods.

---

## [Decision Letter · Decision Letter 1]

27 Jan 2023

Physiological demands and motion analysis of elite foil fencing

PONE-D-22-26762R1

Dear Dr. Bottoms,

We’re pleased to inform you that your manuscript has been judged scientifically suitable for publication and will be formally accepted for publication once it meets all outstanding technical requirements.

Kind regards,

Emiliano Cè

Academic Editor

PLOS ONE

Additional Editor Comments (optional):

Reviewers' comments:

Reviewer's Responses to Questions

**Comments to the Author**

1. If the authors have adequately addressed your comments raised in a previous round of review and you feel that this manuscript is now acceptable for publication, you may indicate that here to bypass the “Comments to the Author” section, enter your conflict of interest statement in the “Confidential to Editor” section, and submit your "Accept" recommendation.

Reviewer #1: (No Response)

2. Is the manuscript technically sound, and do the data support the conclusions?

Reviewer #1: (No Response)

3. Has the statistical analysis been performed appropriately and rigorously? 

Reviewer #1: (No Response)

4. Have the authors made all data underlying the findings in their manuscript fully available?

Reviewer #1: (No Response)

5. Is the manuscript presented in an intelligible fashion and written in standard English?

Reviewer #1: (No Response)

6. Review Comments to the Author

Reviewer #1: (No Response)

7. PLOS authors have the option to publish the peer review history of their article (what does this mean?). If published, this will include your full peer review and any attached files.

Reviewer #1: **Yes: **Mark Willems

---

## [Editor Report · Acceptance letter]

5 Feb 2023

PONE-D-22-26762R1 

Physiological demands and motion analysis of elite foil fencing 

Dear Dr. Bottoms:

I'm pleased to inform you that your manuscript has been deemed suitable for publication in PLOS ONE. Congratulations! Your manuscript is now with our production department. 

Kind regards, 

on behalf of

Professor Emiliano Cè 

Academic Editor

PLOS ONE